# A Qualitative Exploration of the Functional, Social, and Emotional Impacts of the COVID-19 Pandemic on People Who Use Drugs

**DOI:** 10.3390/ijerph19159751

**Published:** 2022-08-08

**Authors:** Erin L. Kelly, Megan K. Reed, Kathryn M. Schoenauer, Kelsey Smith, Kristina Scalia-Jackson, Sequoia Kay Hill, Erica Li, Lara Weinstein

**Affiliations:** 1Department of Family and Community Medicine, Sidney Kimmel Medical College, Thomas Jefferson University, Philadelphia, PA 19107, USA; 2Center for Social Medicine and Humanities, Department of Psychiatry, Semel Institute, University of California, Los Angeles, CA 90024, USA; 3Department of Emergency Medicine, Sidney Kimmel Medical College, Thomas Jefferson University, Philadelphia, PA 19107, USA; 4Center for Public Health Initiatives, Perelman School of Medicine, University of Pennsylvania, Philadelphia, PA 19104, USA; 5Project HOME Health Services, Pathways to Housing PA, Philadelphia, PA 19141, USA

**Keywords:** substance use, medication for opioid use disorder, COVID-19, social networks, social isolation, vaccine hesitancy

## Abstract

Since 2020, people who use drugs (PWUD) experienced heightened risks related to drug supply disruptions, contamination, overdose, social isolation, and increased stress. This study explored how the lives of PWUD changed in Philadelphia over a one-year period. Using semi-structured interviews with 20 participants in a Housing First, low-barrier medication for opioid use (MOUD) program in Philadelphia, the effects of the first year of the COVID-19 pandemic on the daily lives, resources, functioning, substance use, and treatment of PWUD were explored. Interviews were analyzed using a combination of directed and conventional content analysis. Six overarching themes emerged during data analysis: (1) response to the pandemic; (2) access to MOUD and support services; (3) substance use; (4) impacts on mental health, physical health, and daily functioning; (5) social network impacts; and (6) fulfillment of basic needs. Participants reported disruptions in every domain of life, challenges meeting their basic needs, and elevated risk for adverse events. MOUD service providers offset some risks and provided material supports, treatment, social interaction, and emotional support. These results highlight how there were significant disruptions to the lives of PWUD during the first year of the COVID-19 pandemic and identified critical areas for future intervention and policies.

## 1. Introduction

People who use drugs (PWUD), such as opioids, are at elevated risk for severe COVID-19 and mortality, as they have high rates of underlying health conditions [1,2], high rates of homelessness, incarceration, provider-related stigma, and lack of access to health resources [3]. PWUD were also particularly vulnerable to other adverse experiences during COVID-19 due to criminalization of substance use, poverty, unemployment, homelessness, contamination of the drug supply, and risk of overdose [4,5]. During the first year of the pandemic, rates of overdose and medication for opioid use (MOUD) service use changed over time, but ultimately increased, resulting in over 100,000 drug overdose deaths in the US in the 12 month period ending April 2021 [6,7,8,9,10]. Thus far, there is limited literature on the effects for PWUD beyond their rates of infection with COVID-19, use of services, substance use, and rates of overdose. The few qualitative studies that explored the daily experiences, social, and mental health effects of PWUD were conducted in Canada and the United Kingdom [11,12,13,14,15,16], and it is unclear how well these experiences reflect those in the United States.

In addition to national efforts to promote safety during the pandemic—such as mask wearing policies and social distancing guidelines—federal, state, and local policy changes were rapidly enacted during the pandemic to adapt care for PWUD to mitigate disruptions in MOUD services and to preserve housing for vulnerable persons [17]. The effectiveness of these policy changes has largely been positive. Several studies have reported on how adaptations during the pandemic—such as mobile vans with extended release MOUD [18], expanded methadone take-homes [19,20], and telehealth [21]—helped to minimize disruptions to those previously engaged in services. However, many individuals struggled with MOUD adherence during the pandemic and some forms of support were disrupted, such as group therapy visits [22] and harm reduction services (necessitating reuse of injection equipment, which increased risk for COVID-19 and other infections) [23,24]. In addition, PWUD have low vaccination rates for all diseases [25]. Understanding vaccine perspectives specific to PWUD can better support these communities with education and outreach campaigns. Lastly, it is also less clear how well PWUD were able to rally individual-level (e.g., coping ability) and community-level resources for harm reduction practices (e.g., MOUD services, fentanyl testing strips) to mitigate their risk of substance use, social isolation, and safety concerns. Understanding the cumulative effects of the pandemic on all these domains is critical to identifying areas for future interventions as substance use experts anticipated many increased risks for harmful outcomes related to substance use for this population [26]. 

For the present study, we explored how individuals with opioid use disorders (OUD) in an integrated primary and behavioral healthcare, MOUD, and supportive housing program perceived the pandemic’s effects on their lives during the first year of COVID-19 as well as their attitudes towards vaccination, social distancing, and other COVID-19 prevention measures. We sought insight into how policy changes designed to preserve MOUD access and social distancing operated and how social, economic, housing, and community-level stressors contributed to individuals’ ability to engage in harm reduction, engage in MOUD services, and affected their mental health, social connections, resources, and substance use. 

## 2. Materials and Methods

### 2.1. Setting and Study Sample

The structure of the overall low-barrier, integrated MOUD program at the Project HOME Health Services (PHHS) federally qualified health center (FQHC) was described previously [27]. The Integrated Care Clinic (ICC) where this project took place is a partnership between the PHHS FQHC and Pathways to Housing PA (PTHPA), a Housing First program that provides supportive housing to people with serious mental illnesses and substance use disorders [27,28]. The ICC is unique in that the primary care and MOUD services are embedded and integrated with the PTHPA permanent supportive housing services including care coordination, nurse care management, on-site psychiatric services, peer support, and an array of additional services. 

From April 2020 to March 2021, PTHPA served 235 clients with MOUD services, which were predominantly male (59%) and non-Hispanic White (54%). The mean age of those served was 42.3 years old (*SD* = 9.5; range: 24–66). A convenience sample of 20 participants in the ICC MOUD program were recruited by staff at PTHPA. The sample did not significantly differ in the proportion that were male (*z* = −0.51, *p* = 0.61) or non-Hispanic White (*z* = −0.51, *p* = 0.61) of the total population served. Twenty of 23 clients who were approached agreed to participate. Inclusion criteria required participants to be at least 18 years old, English-speaking, currently receiving MOUD services through PTHPA, and able to consent. Study personnel explained the study to interested participants and obtained verbal consent. Verbal consent was obtained to minimize risk of infection during interviews and to protect the anonymity of the participants. 

### 2.2. Procedures

Two female PhD interviewers with extensive qualitative interview experience with PWUD (ELK, MKR) conducted in-depth semi-structured qualitative interviews with participants over the phone or through Zoom from January to April 2021 (see Appendix A for interview guide). The research team collaboratively developed a set of open-ended questions that asked participants about the impacts of COVID-19 on their lives with a focus on social determinants (e.g., food, income, housing, safety, transportation), services, and social impacts. Interviews lasted 10–45 min and participants were compensated with USD 20 cash and two transportation passes for their time. A pseudonym is used when participants are directly quoted. The study was approved by the Thomas Jefferson University Institutional Review Board under protocol no. 20G.344. 

### 2.3. Analysis

A professional transcription service transcribed audio-recorded interviews verbatim. An initial codebook was created by the coding team through line-by-line analysis of three randomly selected interviews and revised according to team consensus. Once a codebook was developed, transcripts were analyzed in Nvivo 12 through directed and conventional content analysis [29]. Interviews were independently coded by four female coders with extensive field experience with PWUD and graduate training in psychology (Ph.D.) and public health (Ph.D. and MPH) as well as undergraduate training in anthropology (BA). Codes with a Cohen’s Kappa (k) below 0.70 were individually reviewed and reconciled. Once intercoder agreement was consistent, alternating pairs of primary and secondary coders independently coded the interviews and any remaining coding discrepancies were resolved via consensus agreement. After completing coding, six authors (ELK, MPH, KMS, KSJ, SH, and KS) composed thematic memos and the group iteratively examined the relationships between the codes to arrive at the central themes. 

## 3. Results

The majority of participants were non-Hispanic White (60%) and male (65%; *n* = 20). The average age was 44.2 years old (*SD* = 8.1; range: 31 to 59 years old) with participants reporting an average of 27 months in MOUD treatment at PTHPA (range: 0.5 to 132 months; see Table 1). Knowledge of personal infection with COVID-19 was rare, with only two participants reporting possible infection. Eight participants discussed knowing others who were infected, and 25% knew someone who died from COVID-19. Six main themes related to COVID-19 were identified during analysis: responses to the pandemic; access to MOUD and support services; substance use; impacts on mental health, physical health and functioning; impacts on social networks; and fulfillment of basic needs. 

### 3.1. Responses to the COVID-19 Pandemic 

#### 3.1.1. COVID-19 Attitudes and Behaviors

Attitudes about COVID-19 (*n* = 13) ranged widely, with some indicating a great of skepticism about how dangerous it was, some reporting apathy, and some a great deal of fear. Of those who were skeptical, two cited conspiracy theories around population control, government sources of COVID-19, or a belief that fentanyl exposure was protective against COVID-19. Among those expressing fear, concerns were wide ranging. These included a general sense of fear from others around them, concern about where they might be at high risk to become infected by COVID-19 (jail or public transit), general fear of what the future held, difficulty getting healthcare from hospitals due to fear of infection, and fears of COVID-19 generally. However, fear of COVID-19 was far from the main stressor for many, as illustrated by Jerry, who said, “I’m more worried about getting shot around here than I am about COVID”.

Personal infection with COVID-19 was relatively rare, as only two assumed they were infected with COVID-19 at one point. However, eight participants discussed knowing others who were infected, and a quarter knew someone who died from COVID-19 infection. Despite varying attitudes about COVID-19, 18 of 20 participants reported wearing a mask at least some of the time and 16 described practicing social distancing. About a third indicated taking COVID-19 precautions very seriously in terms of handwashing, social distancing, and consistent mask use. Vigilance about preventing COVID-19 infection was difficult and took a social and mental toll on people. The majority of participants reported getting tested for COVID-19 at some point, often multiple times through PTHPA, in jail, at the hospital, or at church. Some discussed how they had changed their behaviors to reduce their likelihood of arrest and exposure to COVID-19 because they were afraid of how vulnerable they would be while sharing a cell, as noted by one older male participant (Walter):

Yeah, because jail is even worse, they got people piled up on each other and it takes two weeks to get the results back so you could be bunking with someone who has COVID.

People also perceived a great deal of tension and fear from others when in the community, which exacerbated some of their own negative feelings. A few expressed extreme frustration when others did not comply with COVID-19 prevention measures. Conversely, a small number expressed a great deal of discomfort while wearing a mask or frustration that masks interfered with their ability to read others’ social cues or to make connections to new people or clinical staff. People who disliked masks discussed how it affected them physically and mentally. Mentally, they felt more irritable or felt that others around them were irritated by wearing masks and physically it was hot, uncomfortable, and hard to breathe. Ben noted how COVID-19 guidelines had major personal costs as they interfered with his access to critical resources.

I’ve got it on now. I wear a mask. I stay the distance. I practice the distancing. And mostly now, I stay inside now. But a lot of places I wasn’t able to go. Because of COVID, you can’t get in a lot of places. Even some shelters were limited because of the COVID. A lot of things were limited due to the COVID and the amount of people allowed into these places. So a few places I couldn’t get into and I just had to suffer, I guess, or get used to it.

#### 3.1.2. Vaccination

Among the 19 participants who responded about vaccination, 11 were vaccine-willing, 5 were not vaccine-willing, and 3 were vaccine hesitant. Two participants among the 11 vaccine-willing had already received vaccinations at the time of the interview. The 11 people who were vaccine-willing expressed motivations such as not wanting to be infected with COVID-19 and fears of serious illness or death from COVID-19. When asked if they would hypothetically be willing to be vaccinated that same day, most immediately replied in the affirmative. Some expressed concerns about vaccine side effects; however, these worries did not impact vaccination intent. Gabby said:

Because I want to avoid the virus. I’ll have it protect me against it … but I would still wear my mask and everything, just to be on the safe side.

Others reported previous vaccine hesitancy but ultimately decided in favor of vaccination. Walter said:

I wasn’t going to get it, but I think I’m going to get it now because I just don’t want to take the opportunity of having a third breakout or a big boom, back into red being a high risk.

Three participants were vaccine-hesitant. While two were unsure whether they would ultimately be vaccinated, one believed she would likely get vaccinated after she was able to “see how it’s going” with others. These participants cited deeper concerns about side effects and believed the precautions they were already taking (e.g., handwashing, social distancing) were adequate. Five participants were unwilling to receive a vaccine at the time of the interview. Reasons for refusal included: not wanting to introduce the vaccine substance into their body, previous adverse experience with influenza vaccination, government distrust, and beliefs that their body’s immune system response was adequate. Two noted a belief in conspiracy theories related to the COVID-19 vaccine, fertility, and population control. Bob was adamant about his decision to remain unvaccinated:

You’ve got it or you don’t. You’re going to get it or you won’t. You got it, not being around people that have it, period. That’s all there is to it.

### 3.2. Connection to MOUD Services

One year into the COVID-19 pandemic, PTHPA was a main service provider for all participants, though only five exclusively received services there. Participants described receiving telehealth services, home visits, as well as regular clinic visits throughout the pandemic, albeit less frequently for some (*n* = 5). Participants largely reported core MOUD services were accessible during COVID-19 with six individuals stating directly that there was no change in their service access or quality. One participant noted that despite a decrease in the number of staff, all the normally scheduled home visits, transportation, clinic visits, and money distribution did not change in function. “Anything I need help with, they are there for me”, said one participant (Ben) and another commented that PTHPA was his lifeline. A few participants had concerns about PTHPA staff turnover as interfering with rapport and one client expressed concern that PTHPA was trying to help too many high-need people simultaneously, which limited their capacity to help her. Five interviewees noted the loss of MOUD group structure as an important place for social interaction, an activity replacement for substance use, and a reason to come into the office more routinely for services. The loss of these groups increased their sense of isolation and anxiety.

The majority of participants (*n* = 13) also described receiving services from other specialty substance use treatment centers including outreach, harm reduction, housing, inpatient rehabilitation, methadone treatment, and abstinence-based treatment groups (e.g., Narcotics Anonymous). Many reported some disruptions or adaptations by these providers during the pandemic, such as reduced bed capacity, online groups, and lower capacity for in-person treatment groups. All participants receiving outside methadone treatment (*n* = 4) reported that clinic resources were operating at full capacity with some modifications (such as conversion of Narcotics Anonymous meetings from in-person to Zoom).

### 3.3. Substance Use, Harm Reduction, and Overdose

#### 3.3.1. Substance Use

We did not ask people for a comprehensive substance use history, only about the frequency and quantity of their use in the prior year. Among those reporting a decrease, they attributed it to social-distancing and staying home alone, fear of incarceration and subsequent infection, and loss of income. Tag outlined how access to his MOUD helped to facilitate his staying home, as it decreased his need to seek out drugs.

It’s been affected. I don’t go out as much, so I just use my Suboxone. It is helpful in a way because I won’t go out as much to gets drugs. I’ll use my Suboxone. So in a way it’s helpful.

Six others described decreased use over the prior year due to personal motivation, positive buprenorphine/methadone experiences, and changes in behavior following adverse experiences and increased access to mobile services. Those who said COVID-19 directly increased their substance use (*n* = 2) explained that being forced to stay home caused boredom that increased their cravings and use.

Of the 16 participants that discussed the local drug supply, the majority reported it had worsened during COVID-19 (*n* = 11) and several reported contamination of all substances. Participants described heroin suspected as adulterated with fentanyl, xylazine (a veterinary tranquilizer), or amphetamines and of powder or crack cocaine adulterated with fentanyl or unknown substances. These participants expressed a lack of confidence in the identity and purity of their substances of choice and fears of unexpected, adverse physical reactions of vomiting, bacterial infections, and overdose. ClaireBear expressed her deep frustration with the changes to the drug supply and its impact on her health.

They cut it with crap. It’s not like heroin...Yeah, it’s tranquilizer, which affects your whole GI system. It’s not even for human consumption, and then your GI system doesn’t work for a couple days so that anything you eat goes putrid in your stomach. It comes up either through your mouth or out the other way, and it’s horrible.

Three participants explicitly expressed that their frequency/quantity of substance use was directly impacted by the contaminated drug supply. Five individuals reported no change in the cost, availability, nor quality of the drug supply during COVID-19, which they attributed to their long-standing relationships with their suppliers. While they did not express personal fear of contamination of their drugs, they acknowledged that the purity of the general supply fluctuated throughout the city.

#### 3.3.2. Harm Reduction

All participants named at least one method of harm reduction that they employed while using substances during the pandemic, namely MOUD services (*n* = 19). For the 16 who still used drugs at some point in the prior year, most still used in the presence of another person or asked someone to check in on them (*n* = 10), and some utilized fentanyl test strips (*n* = 6) and naloxone (*n* = 5). Eight participants described partaking in actions that increased their agency over the frequency of their use and methods of consumption (i.e., not selling their buprenorphine, spacing out doses, sniffing/snorting instead of injecting, avoiding triggering spaces/people, not sharing syringes). Access to naloxone was seen as plentiful during the pandemic, as Bob described, “I got more Narcan than the whole world, and I’ve got neighbors”. Several participants also remarked that their consistent access to naloxone was critical to reversing witnessed overdoses.

#### 3.3.3. Overdose

Four participants discussed personal overdose or extreme withdrawal that occurred during the pandemic. Participants recalled their experiences within the context of other complex vulnerabilities, including the transition in and out of incarceration, mental health crises, social isolation, and termination of methadone treatment. Almost half of the participants reported knowing people who overdosed in the prior year, such as a friend’s overdose described by Elisha.

My one friend, he had went down and, and he got his drugs and he went to use in the usual place that he does where they’re all other people and the cops, they come and told everyone they, they couldn’t be in groups anymore. So they made everyone disperse, which caused him to, you know, get on the train and use on the train and then he used too much. And you know, overdose.

### 3.4. Impacts on Mental Health, Physical Health, and Functioning

The mental health effects of the pandemic were highly individualized. Several participants expressed a great deal of fear about the pandemic, but others reported skepticism and apathy. Of those who were skeptical, two cited conspiracy theories around population control, government sources of COVID-19, or a belief that fentanyl exposure was protective against COVID-19. COVID-19-specific fears broadly fell into three categories: (1) a general sense of fear from being around others; (2) concern about high risk of infection COVID-19 in jail, public transit and hospital settings; and (3) fear for the future. Vigilance about preventing infection was difficult and took a social and mental toll on participants.

When directly asked about their mental health, almost half initially reported no adverse effects and two reported improvement. Importantly, most participants who had mental health struggles reported that they could turn to PTHPA for support. However, later in the interview, eight said that they have felt more lonely, isolated, irritable, or anxious. These impacts were largely attributed to non-pandemic issues, such as housing quality, deaths, and substance use. The pressures of the pandemic also raised some existential questions and many participants described a complex, evolving reaction over time, which was best described by Miss B:

Being under these conditions I would say, and I’m speaking in general but I’m also speaking for other people, it kind of forces you to look at your life. It forces you to face reality and figure out what you want to do and what it’s going to be like because the COVID came on so strong and fast and for all those people who died, it’s caused me to feel depressed a lot. It’s caused me to realize how little of a life I have. How empty and it’s just the reality of it. It’s what it is. It’s just crazy. It brings fear for the future. You never know what’s going to happen and you feel like... I feel like I’ve had to cherish every little thing. I’ve also felt offended a lot. There’s nothing I can do about it. It’s going to be what it’s going to be.

The majority of participants (*n* = 11) stated their physical health was “okay” or “good”, though several described complex histories of hospitalizations for overdose, injuries, and other health conditions and six reported their health was adversely affected by COVID-19. Those who reported their health as “bad” or “not good” typically reported mobility issues, such as back or leg pain, that interfered with daily tasks. Almost half reported receiving assistance with their physical health from PHHS and PTHPA during the pandemic and there was little note of disruptions to access to primary care services.

Most individuals (*n* = 12) reported COVID-19 impacted their daily routines. Of those limiting their daily activities by staying at home, most reported feelings of isolation, irritability, anxiety, or loneliness related to their restricted activities. For example, Savannah described how shut downs due to the pandemic closed off many of her routines

Let’s see. It’s hard to, let’s see. It’s a lot harder shopping and getting everything that I need because a lot of places aren’t even open. You can’t really go out and eat anywhere. You know how you go out and you have conversations? You know what I mean? You can’t do all that, so yeah. It’s a big blocker, also.

However, three continued their daily activities normally, enjoying a mix of indoor and outdoor activities. Some reported going to PTHPA as a reason to leave their homes and for needed social interaction.

### 3.5. Impact on Social Connections

Several participants reported missing social activities (e.g., family gatherings, going to groups) as the vast majority (*n* = 16) reported social distancing efforts. Throughout the interviews participants described impacts on their relationships and all were asked to identify the three people closest to them in their social networks (SNs). The majority (*n* = 12) identified three people that they were close to, primarily family members, and half reported daily contact with someone. Many reported less frequent contact with family, particularly in-person contact due to distance, infection concerns, or transportation issues. This was particularly difficult for the loss of contact with children, as described by Elijah,

It’s not like it was before where we’re seeing each other every day and, or like every week, especially with the kids and in COVID, you know, it’s we don’t want to have them on the bus or like on the train or whatever, any type of transportation we get over here and vice versa. So, we’ve kind of just been doing it over the phone…but it’s just not the same as being there in person.

A few described still seeing their closest friends and family on a daily basis and feeling closer to them as the pandemic meant they spent more quality time together. For two people, their only contacts were with a service provider and another two were only seeing their social network members via the phone and they reported feeling lonely, anxious, and depressed. A quarter described the importance of social connections with service providers and other patients while they lacked access to their closest friends and family.

Loss was a significant theme. Almost half of participants reported a loss during the pandemic, and the majority of those deaths were due to COVID-19 (5 out of 9). One interviewee spoke around the abrupt loss of her friend, who went into the hospital after feeling unwell and passed away of COVID-19 shortly after, stating “…and my heart is broken to the core”. Participants also lost friends, spouses, or family members to other causes, including drug use.

### 3.6. Attendance to Basic Needs

At the time of interview, 17 participants were housed and three did not have stable housing (in a shelter, in a hotel, and homeless/”couch surfing”). Only two interviewees discussed housing changes during COVID-19. Savannah said she took an undesirable apartment out of fear that there would not be any options available in the future due to COVID-19 and Walter explained that couch-surfing was more difficult now due to social distancing. While the majority were satisfied with their living conditions, some reported wanting a change due to noisy neighbors, bug infestation, and community violence (*n* = 5).

Access to food varied and even among those who said they had enough food (*n* = 11) four indicated that they would run out at the end of the month. This required individuals to be vigilant for community-level resources, which Ben found to be plentiful if a person knew where to look.

There’s people who will pull up from anywhere at any time with food. I’m going to tell you something nice. There’s so many people that give away food, cook food and come down there and give it out, I swear to God. I’m not kidding you. They do have food services down there. And they have Prevention Point, the community center, even though I said you can’t stay in there during COVID, they still have coffee in the morning and they’ll have donations from Wawa. They’ll hook you up. And then also, on the other side of Huntingdon, they have St. Francis. And St. Francis has a meal every night for you.

For some, their circumstances improved, as they noted their benefits increased during the pandemic (*n* = 6). There was a near even split in access to increased formal benefits (*n* = 11 receiving), such as supplemental security income and stimulus checks, though two reported difficulty accessing these funds. Informal means of income, such as odd jobs or panhandling, diminished during the pandemic due to social distancing (*n* = 8) and several stated that they could not meet their daily needs (*n* = 7) or lacked basic supplies at the end of the month (*n* = 13), described succinctly by Scott as “I don’t even have toilet paper for my ass, or food”. Several participants described new barriers to transportation during the pandemic, as they reported fewer buses and trains in operation with decreased hours and three described witnessing increased violence on public transit. Several participants (*n* = 6) decreased travel due to anxiety about COVID-19, having less places to go, and a lack of funds for transportation. Over half of participants reported that their access to transportation did not change during COVID-19 or increased, largely due to PTHPA still picking them up for their healthcare appointments.

## 4. Discussion

The present study explored the intersection of the ongoing opioid overdose crisis and the first year of the COVID-19 pandemic on PWUD in a supportive housing setting and resulted in six identified themes: response to the COVID-19 pandemic, access to MOUD and support services, substance use, impacts on mental health, physical health and functioning, impacts on social networks, and fulfillment of basic needs. Remarkably, participants in the present study reported little MOUD service disruption almost one year into the pandemic, which suggests that the policy changes regarding MOUD (e.g., easier access to telehealth, allowance of telephone services, allotment of greater take-home doses) and other supports (e.g., stimulus payments, increased benefits) supported access to and continuity of MOUD. However, some important services, such as group meetings, decreased during the pandemic, and were keenly missed. Across states, MOUD services connections varied highly during a comparable timeframe. A Pennsylvania study analyzing claims from January to October in both 2019 and 2020 found that the onset of the COVID-19 pandemic was associated with a reduction in both the number of new patients initiating MOUD treatment and the number of patients filling buprenorphine prescriptions [30]. In contrast, a Texas study utilizing the Prescription Drug Monitoring Program (PDMP) found a marked increase in new buprenorphine patients in the first 90 days of the pandemic [31]. It is possible that the low-barrier program model in the present study allowed for easier retention and future research should explore what specific mechanisms were critical to MOUD retention and should be perpetuated.

Global and local disruptions to the unregulated drug market during this time were associated with elevated risk of harm, such as overdose. Accordingly, several studies noted increased drug use by some PWUD [11,13], as well as changes to MOUD service capacity [12]. In our sample, changes to the drug supply meant that participants often felt that there were higher rates of adulterants, such as fentanyl or xylazine, which made drugs more dangerous overall. Only two participants reported increased drug use, perhaps reflecting the ongoing connection to MOUD services by participants. Despite personal use being stable or reduced, participants still contended with several adverse events related to drug use, including personal or witnessed overdose, impaired health, decreased access to inpatient rehabilitation services, and impaired relationships. Some harm reduction strategies, such as using with others, were more difficult but not absent among participants. The strategies that were less disrupted included access to their MOUD medication, having naloxone on hand, and fentanyl test strips, which suggest that local policies were effective at providing resources and tools among participants connected to care.

As has been found previously among PWUD [32], some participants reported increased feelings of sadness, boredom, irritability, anxiety, and loneliness during the pandemic. Participants described how they were negatively affected by their compliance with social distancing and restricted activities in the general community and the changes to their social environment were often directly referenced as causing their mental health distress. While there were relatively low rates of personal infection among our participants, several had lost a friend or family member to COVID-19, exacting a significant emotional toll. The majority of individuals were able to maintain some contact with friends and family in-person, albeit decreased due to poor phone access and lost abilities to travel to see family (due to insufficient funds or social distancing). Few participants described total social isolation but the frustrations of their social constrictions within the broader community and beyond their network were a pervasive challenge. Nationally, reports of mental health symptoms and poor access to mental health services increased over the course of the pandemic [33,34,35] and future research should explore the impacts on those in public services in order to determine where additional resources are most urgently required. Many credited their service providers as providing essential emotional and instrumental social supports, particularly among those who lacked any contact or had especially reduced access to their closest friends and family during COVID-19. Encouragingly, many participants described retaining their access to physical health services, as they received their MOUD through a specialized primary care designed for PWUD, which highlights the value of integrated care programs to address the needs of highly vulnerable populations.

Participants in this study were mixed in their intentions to receive the COVID-19 vaccine, though most intended to be vaccinated or had already received their COVID-19 vaccination by the start of the interview. These findings complement those of an Australian survey with PWUD where most indicated they would be vaccinated, but 15% said they would definitely not [36]. Further survey research with this cohort confirmed that vaccine rates for PWUD lagged behind those of the general population [37]. Concerns among participants were like those of other Americans with varying levels of medical mistrust. General medical mistrust and beliefs regarding COVID-19 include concerns that the government cannot be trusted to tell the truth about COVID-19, information is being held back by the government, and anxieties about the safety and efficacy of the COVID-19 vaccine [38]. In many ways, this suggests that while PWUD should receive tailored risk communication about the COVID vaccine, the content can remain similar to that delivered to other groups with some additional considerations. For example, PWUD may have suppressed immune systems due to HIV infection [39] and inadequate nutrition and hydration, especially during periods of heavier drug use [40]. This increased susceptibility to morbidity from COVID-19 should be communicated in a non-judgmental manner. The rewards of vaccination should be emphasized, as many participants indicated their mental health had worsened during the COVID-19 pandemic. Vaccination would allow PWUD the opportunity to reconnect with family and friends and to feel less nervous when in crowded places or among others. Overall, these findings highlight the need for increased public mental health efforts to help rebuild social connections in communities particularly affected by opioid use and COVID-19, as they experienced significant upheaval and urgently need intervention.

Participants in this supportive scattered site housing program maintained housing throughout the first year of the pandemic, which may reflect both the effectiveness of the housing model as well as federal policies that prevented eviction. Certain segments of PWUD communities are at increased risk of COVID-19 infection due to congregate living situations such as recovery houses and homeless shelters [3]. Emerging research indicates that messages focused on altruistic appeals are effective at convincing vaccine-hesitant people to accept the COVID-19 vaccine [41]. This approach may translate well with PWUD. PWUD often have limited access or interrupted access to cellphones and the internet [42], which means that education about vaccines needs to occur where PWUD spend time. Community-based organizations already providing services to this population are natural partners for vaccination programs. PWUD are open to discussions about COVID-19 vaccination and have similar motivations and concerns about vaccinations as people who do not use drugs. Like other marginalized populations, messages—and vaccinations—should be brought to locations where PWUD frequently visit. These include harm reduction services such as syringe exchange programs, programs where people receive MOUD, housing programs, transit hubs, food banks and related food-access services, libraries, and mutual aid programs such as 12-step programs.

Similar to other studies with PWUD [12], many participants in the present study lacked necessary resources, such as enough food or basic necessities. Encouragingly, almost half reported accessing their primary care services and had access to mental health support services. Participants described multiple concurrent stressors related to housing or neighborhood issues, lost or reduced opportunities for employment, food scarcity, boredom, loneliness, sadness, irritability, and anxiety. However, they often did not directly attribute their personal issues to the pandemic, perhaps reflecting the chronic distress present in their lives. In particular, increasing neighborhood gun violence was a significant source of stress for these participants, as rates of gunshot deaths in Philadelphia increased dramatically in 2020 and in 2021 [43].

### Strengths and Limitations

Among the strengths of this study was the comprehensive assessment of how the social determinants of health were affected for this population, the use of social network mapping interviewing, access to resources and supports across agencies and systems, attitudes and experiences with COVID-19, as well as probing about the local drug supply. Despite these strengths, the study is limited by the small sample size that was drawn from one program in one city. However, our participants reflected the experiences of people at a range of different points in their recovery, did not differ in terms of gender or race/ethnicity from those receiving services from the clinic during that period, and were drawn from Philadelphia, a major center for the opioid crisis, meaning that concerns about their representativeness should be somewhat mitigated. Most participants were housed and many engaged in at least some social distancing practices within their living spaces, which may not reflect the experiences of many PWUD. Participants were already accessing services from one organization, which may contribute to the high use we found of other services and high access to harm reduction tools. Similarly, participants expressed relatively stable access to services and satisfaction with services they received. This may be somewhat unique to the low threshold services received and the COVID-19 responses of those agencies. Finally, interviews were subject to social desirability bias. Future studies might expand on these findings with a larger sample in a broader range of contexts and service providers to establish their generalizability.

## 5. Conclusions

MOUD service access and retention were critical goals during the pandemic to protect highly vulnerable PWUD. For those with access, Housing First services played a crucial role in maintaining engagement in MOUD services and avoidance of harm and expansion of these services could help PWUD who remain vulnerable. Contrary to national trends, the majority of our sample were stable or able to reduce their use of substances, which highlights the critical role of low-barrier MOUD services during the first year of the pandemic. The relaxation of the restrictions related to the prescription of buprenorphine and methadone helped to preserve the continuity of care for PWUD and policy makers should consider making these changes permanent to improve treatment access in the future.

## Figures and Tables

**Table 1 ijerph-19-09751-t001:** Sample demographic and clinical characteristics (*n* = 20).

		*n*	*%*
Gender			
	Cisgender man	13	65%
	Cisgender woman	7	35%

Race/Ethnicity			
	White	12	60%
	Black	4	20%
	Latinx	3	15%
	Other	1	5%

		* M *	* SD *
Age		44.2	8.1

Number of Months in MOUD Treatment		27.3	29.6

## Data Availability

Data is available through requests to Lara Weinstein after review.

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
