# Peer review of "A Qualitative Exploration of the Functional, Social, and Emotional Impacts of the COVID-19 Pandemic on People Who Use Drugs"

_ijerph, 2022, doi:10.3390/ijerph19159751_

Round 1

Reviewer 1 Report

In this work, the authors interviewed people who use drugs (PWUD) who also access a Housing First, low-barrier medication for opioid use program in Philadelphia about their daily lives, resources, functioning, substance use, and treatment were impacted during the first year of the COVID-19 pandemic. Results indicate that engagement with the Housing First, low-barrier medication for opioid use program might be a protective against negative outcomes from or exacerbation of existing substance use disorder and other daily stressors experienced by PWUD. This work is an important perspective on the far-reaching positive impact these programs offer to those who have access to and engage with them. 

Generally: this work would benefit from a careful review focusing on clarity, grammar, spelling, and uniform writing style.

Introduction: content is somewhat non-linear, transition between paragraph 2 and 3 would aid the reader in understanding the purpose of the work and necessary background information.

Section 1, line 47: mention of "service use" but no definition of what "service use" refers to. 

Section 1, line 51/52: "The few qualitative..." sentence is awkward and does not make sense.

Section 1, line 58: MOUD introduced without definition.

Section 1, lines 69-71: "These outcomes are critical..." this sentence is awkward and confusing.

Section 1, line 79: OUD introduced without definition.

Section 2.1, line 90: "fenderally-qualified" typo?

Section 2.1, line 101: were there any exclusion criteria?

Section 2.1, line 102: only verbal consent was obtained, perhaps mention that this was done to protect the identities of the respondents.

Section 2.2, line 108: "The research team collaboratively..." this sentence is missing a word, confusing. 

Section 2.2: Will the questions used in this work be made available via SI or other online repository?

Section 2.2, line 109: social determinants used here without definition. Perhaps in the introduction the "social determinants" considered in this work could be defined.

Section 2.2, line 112: please list the IRB approval number.

Section 2.3, line 119: what is the relevance of listing the sex of the researchers?

Section 3.1, line 130: list the range of ages alongside the mean and SD.

Section 3.1, Table 1: How do the demographics of the convenience sample considered in this work relate to the demographics of the population historically served by the PTHPA?

Section 3.2.1, line 165: Reconsider the alias "Whop" or clarify that it represents a person when first introduced. As written, it is confusing.

Section 3.4.1, line 246: The attribution of quotes has suddenly changed. Please use uniform writing styles so the author can follow along more easily.

Section 3.5, line 298: "COVID-19 based fears included..." sentence appears to be missing a word, is confusing as written.

Section 4.1, line 407: "At both the state and national..." sentence is confusing and its purpose is unclear.

Section 4.1, line 484: "PWUD often have limited access..." sentence is awkward.

Section 4.2: How did the use of a convenience sample impact the data gathered in this work? What is the power of this work given the sample size and method of participant identification? 

Author Response

In this work, the authors interviewed people who use drugs (PWUD) who also access a Housing First, low-barrier medication for opioid use program in Philadelphia about their daily lives, resources, functioning, substance use, and treatment were impacted during the first year of the COVID-19 pandemic. Results indicate that engagement with the Housing First, low-barrier medication for opioid use program might be a protective against negative outcomes from or exacerbation of existing substance use disorder and other daily stressors experienced by PWUD. This work is an important perspective on the far-reaching positive impact these programs offer to those who have access to and engage with them. 

We thank the reviewer for their kind words. We have addressed each of the issues raised below. 

Generally: this work would benefit from a careful review focusing on clarity, grammar, spelling, and uniform writing style.

We made edits throughout the manuscript to improve its legibility. 

Introduction: content is somewhat non-linear, the transition between paragraph 2 and 3 would aid the reader in understanding the purpose of the work and necessary background information.

We agree that the transition between paragraphs 2 and 3 required editing and we have edited this section for readability.

Section 1, line 47: mention of "service use" but no definition of what "service use" refers to. 

We clarified that in this instance, we are referring to MOUD service use.

Section 1, line 51/52: "The few qualitative..." sentence is awkward and does not make sense.

We edited for clarity.

Section 1, line 58: MOUD introduced without definition.

The acronym was explained in the abstract but it has been added to the main text for clarity.

Section 1, lines 69-71: "These outcomes are critical..." this sentence is awkward and confusing.

We edited this sentence for clarity.

Section 1, line 79: OUD introduced without definition.

We clarified the acronym.

Section 2.1, line 90: "fenderally-qualified" typo?

Yes- we meant to say federally-qualified

Section 2.1, line 101: were there any exclusion criteria?

The exclusion criteria mirrored the inclusion criteria and therefore exclusion criteria were not added for purposes of brevity.

Section 2.1, line 102: only verbal consent was obtained, perhaps mention that this was done to protect the identities of the respondents.

In addition to protecting the identities of the participants, it was not possible to meet with participants for interviews during the pandemic due to concerns about infection, which necessitated the conduct of interviews over telephone or Zoom for participant safety. This information has been included in the paper.

Section 2.2, line 108: "The research team collaboratively..." this sentence is missing a word, confusing. 

We removed the extra preposition that was making the sentence confusing.

Section 2.2: Will the questions used in this work be made available via SI or other online repository?

We would be happy to share our interview guide as a supplemental index.

Section 2.2, line 109: social determinants used here without definition. Perhaps in the introduction, the "social determinants" considered in this work could be defined.

We added this information to section 2.2.

Section 2.2, line 112: please list the IRB approval number.

The IRB number was added to the text.

Section 2.3, line 119: what is the relevance of listing the sex of the researchers?

This is a required element of the COREQ checklist from the EQUATOR network that establishes the requirements for the quality of qualitative research methods, which we used to guide the elements of our manuscript.

Section 3.1, line 130: list the range of ages alongside the mean and SD.

We added the age range as requested.

Section 3.1, Table 1: How do the demographics of the convenience sample considered in this work relate to the demographics of the population historically served by the PTHPA?

We provided descriptives of the total population served over a one-year period (n = 235) and our sample did not significantly differ in terms of sex or race/ethnicity from the total population served in two-tailed z-score proportion tests. While it may appear that it is a typo that the z score proportions are the same, they are in fact z = .51 and p = .61 for both tests and if we presented the tests out to 4 digits, then the slight differences would become apparent.

Section 3.2.1, line 165: Reconsider the alias "Whop" or clarify that it represents a person when first introduced. As written, it is confusing.

The participants were able to self-select their pseudonyms, which is how that one was selected. We modified this pseudonym to Walter throughout for clarity.

Section 3.4.1, line 246: The attribution of quotes has suddenly changed. Please use uniform writing styles so the author can follow along more easily.

We harmonized how the quotes were attributed throughout the results section.

Section 3.5, line 298: "COVID-19 based fears included..." sentence appears to be missing a word, is confusing as written.

This sentence was edited for clarity.

Section 4.1, line 407: "At both the state and national..." sentence is confusing and its purpose is unclear.

This sentence was edited for clarity to reflect state-level differences.

Section 4.1, line 484: "PWUD often have limited access..." sentence is awkward.

This sentence was edited for clarity.

Section 4.2: How did the use of a convenience sample impact the data gathered in this work? What is the power of this work given the sample size and method of participant identification? 

With the addition of our analyses comparing the demographics of our sample to the total population served by the clinic during the first year of COVID-19, we did not find any significant differences between our sample and all those served. This should mitigate some concerns about how representative our sample was of the community of those in services during this period. However, the goal of qualitative research is to explore the experiences of the participants in a more holistic manner than is possible with quantitative methods and therefore representativeness is an imperfect concept in this context.

Reviewer 2 Report

The study is based on online conduct of interviews. The study is very well executed. However, there are certain points that needs attention: 

1. Please check the sentence in the abstract 'Using semi-24 structured interviews with twenty PWUD participating in a Housing First, low-barrier medication for opioid use (MOUD) program in Philadelphia, we explored how the daily lives, resources, functioning, substance use, and treatment of PWUD were affected during the first year of the 27 COVID-19 pandemic.'

2. 'provider stigma' term may be replacedd with 'provider-based stigma'

3. Please check the sentence grammar like 'and it unclear how well these experiences reflect those in the 53 United States'; 'Population adherence to following COVID-19 protection measures and vaccination 73 as it became available was unclear.'

4. Is verbal consent for the study sufficient enough to carry out the study? Please justify 

5. The sample size is very less. I think the title should be modified accordingly to justify the same

Best wishes 

Author Response

The study is based on online conduct of interviews. The study is very well executed.

We thank the reviewer for their thoughtful attention to our paper and our responses to each of their points is addressed below.

However, there are certain points that need attention: 

  1. Please check the sentence in the abstract 'Using semi-24 structured interviews with twenty PWUD participating in a Housing First, low-barrier medication for opioid use (MOUD) program in Philadelphia, we explored how the daily lives, resources, functioning, substance use, and treatment of PWUD were affected during the first year of the 27 COVID-19 pandemic.'

We edited this sentence. 

  1. 'provider stigma' term may be replaced with 'provider-based stigma'

We edited the term per the request of the reviewer.

  1. Please check the sentence grammar like 'and it unclear how well these experiences reflect those in the 53 United States';

Thank you for pointing out the missed verb.

  1. 'Population adherence to following COVID-19 protection measures and vaccination 73 as it became available was unclear.'

Thank you for pointing out the lack of clarity in this sentence. We edited it for clarity.

  1. Is verbal consent for the study sufficient enough to carry out the study? Please justify 

It was not possible to meet with participants for interviews during the pandemic due to concerns about infection, which necessitated the conduct of interviews over telephone or Zoom for participant safety. All participants were provided with a study information sheet and the trained interviewers were able to answer any questions during the consent process.

  1. The sample size is very less. I think the title should be modified accordingly to justify the same.

In qualitative research studies, there are no clear rules for sample sizes. Rather data is collected to data saturation, as was done in this study. We modified the title to reflect the qualitative nature of the study. As noted in our response to Reviewer#1, we also added information about the total population of those served during the recall period of the study and did not find significant differences between our sample and the total population of those served by the clinic during that period.